Polyp expansion of passive suspension feeders: a red coral case study

Rossi Sergio sergio.rossi@unisalento.it 1 2
Rizzo Lucia 3
Duchêne Jean-Claude 4
1 Dipartimento di Scienze e Tecnologie Biologiche ed Ambientali, Università del Salento , Lecce , Italy
2 Institut de Ciencia i Tecnologia Ambiental, Universitat Autónoma de Barcelona , Cerdanyola del Vallés , Spain
3 Stazione Zoologica Anton Dohrn , Napoli , Italy
4 Station Marine d’Arcachon, Environnements et Paleoenvironnements Océaniques, Université Bordeaux I , Bordeaux , France
Banaszak Anastazia
Electronic publication date: 2019 Jul 9
Publication date: 2019
Volume: 7
Electronic Location ID: e7076
Received 2019 Jan 16; Accepted 2019 May 4
Copyright: ©2019 Rossi et al.
Copyright year: 2019
Copyright holder: Rossi et al.
License: This is an open access article distributed under the terms of the Creative Commons Attribution License, which permits unrestricted use, distribution, reproduction and adaptation in any medium and for any purpose provided that it is properly attributed. For attribution, the original author(s), title, publication source (PeerJ) and either DOI or URL of the article must be cited.
License URL: https://creativecommons.org/licenses/by/4.0/

Keywords: Octocorals, Passive suspension feeders, Optimal foraging theory, Corallium rubrum, Activity rhythms, Trophic ecology, Benthic-pelagic coupling

Funding: Spanish Ministry of Education and Culture PB94-0014-C02-01 The Beca Posdoctoral del Ministerio de Ciencia e Innovación (EX2006-1730, executed in the Prof. Antoine Grémare laboratory) The Ramón y Cajal contract (Ramón yCajal -2007-01327) Marie Curie International Outgoing Fellowship (ANIMAL FOREST HEALTH, project number 327845) Generalitat de Catalunya to the project MERS (2017 Suports Grups de Recerca - 1588) ICTA “Unit of Excellence” (MinECo, MDM2015-0552) Support for this study was provided by a Formación de Personal Investigador fellowship from the Spanish Ministry of Education and Culture, which was granted to Sergio Rossi under the project PB94-0014-C02-01 for the in situ observations. The Beca Posdoctoral del Ministerio de Ciencia e Innovación (EX2006-1730, executed in the Prof. Antoine Grémare laboratory), the Ramón y Cajal contract (Ramón yCajal -2007-01327) and a Marie Curie International Outgoing Fellowship (ANIMAL FOREST HEALTH, project number 327845) were essential for the ex situ experiments and data treatment. Funding was also provided by the Generalitat de Catalunya to the project MERS (2017 Suports Grups de Recerca - 1588). This work contributes to the ICTA “Unit of Excellence” (MinECo, MDM2015-0552). The funders had no role in study design, data collection and analysis, decision to publish, or preparation of the manuscript.

==============================
Polyp activity in passive suspension feeders has been considered to be affected by several environmental factors such as hydrodynamics, water temperature and food concentration. To better elucidate the driving forces controlling polyp expansion in these organisms and the potential role of particle concentration, the octocoral Corallium rubrum was investigated in accordance with two approaches: (1) high-frequency in-situ observations examining various environmental and biological variables affecting the water column, and (2) video-recorded flume-controlled laboratory experiments performed under a range of environmental and biological conditions, in terms of water temperature, flow speed, chemical signals and zooplankton. In the field, C. rubrum polyp expansion correlated positively with particle (seston and zooplankton) concentration and current speed. This observation was confirmed by the flume video records of the laboratory experiments, which showed differences in polyp activity due to changes in temperature and current speed, but especially in response to increasing nutritional stimuli. The maximum activity was observed at the highest level of nutritional stimulus consisting of zooplankton. Zooplankton and water movement appeared to be the main factors controlling polyp expansion. These results suggest that the energy budget of passive suspension feeders (and probably the benthic community as a whole) may rely on their ability to maximise prey capture during food pulses. The latter, which may be described as discontinuous organic matter (dead or alive) input, may be the key to a better understanding of benthic-pelagic coupling processes and trophic impacts on animal forests composed of sessile suspension feeders.

Introduction

Passive suspension feeders play an important role in energy transfer from the water column to the benthos (Gili & Coma, 1998). These organisms are important biomass contributors in benthic communities, being an essential part of the ‘animal forest’, in which the main three-dimensional builders are clonal or individual organisms of animal origin (Rossi et al., 2017). Seston (live and dead particles present in the water column, Rossi & Gili, 2007) availability, which depends on its abundance, composition and renewal rate, is one of the most important parameters affecting the distribution, energy fluxes and biological constraints of suspension feeders (Grémare et al., 1997; Coma et al., 2001). As such, it is key to expanding our knowledge of community dynamics and the potential regression, substitution and/or mortality of suspension feeder populations, processes that have led to profound transformations over the last few decades (Rossi, 2013).

An immediate response of passive suspension feeders to changes in seston availability and hydrodynamism is to tune their feeding activity, which can also be affected by other short-term and seasonal environmental and biological changes (Coma et al., 1994; Rossi & Gili, 2007; Previati et al., 2010). In gorgonians, alcyonarians and zoanthids for example, feeding activity is reflected in polyp expansion (Dai & Lin, 1993). It has been demonstrated to vary seasonally (Coma et al., 1994; Garrabou, 1999; Rossi, 2002), with the frequency of food inputs as potential factor driving polyp expansion (Tsounis et al., 2006; Rossi & Gili, 2007; Rossi, Coppari & Viladrich, 2016).

The more variable the water column environmental factors, the more diversified the mechanisms to enhance prey capture and feeding optimisation, as organisms adopt a range of strategies to take advantage of every potential source of food (Coma et al., 2001). In laboratory experiments, capture rates and polyp expansion among passive suspension feeders have been shown to be related to nutritional stimuli, particle concentration and flow speed (Leversee, 1976; Dai & Lin, 1993; Anthony, 1999), but there is almost no information on how environmental factors affect in situ polyp activity during short-term cycles (Rossi & Gili, 2007). Epibenthic water masses and associated plankton and seston concentrations can change rapidly, with particulate organic matter concentration tripling or quadrupling in less than a day (Grémare et al., 2003; Rossi & Gili, 2007; Rossi et al., 2013). These non-continuous food pulses have never been fully studied in relation to the activities of passive suspension feeders and may be a key factor for understanding the overall energy budget of sessile organisms in animal forests.

The aim of this study is to achieve a better understanding of which factors drive polyp activity in passive suspension feeders, seeking to determine whether environmental and biological factors act synergistically in polyp expansion, using the red coral (Corallium rubrum) as a model organism. To achieve this objective, two methodological approaches were used: (1) high-resolution polyp observations in the field (Rossi & Gili, 2007), under recorded environmental conditions (i.e., current speed, zooplankton concentration, chlorophyll a and protein content of epibenthic seston). This will help to better understand whether there is any pulse-like energy input (Palardy, Grottoli & Matthews, 2006; Tsounis et al., 2006). (2) ex-situ high-resolution flume (closed channel under controlled conditions) experiments to test C. rubrum polyp activity in relation to a range of environmental factors (i.e., temperature, current speed, nutritional stimuli and presence of zooplankton). The main aim is to understand how intermittent food supply may influence the energy budgets of passive suspension feeders.

Materials and Methods

Field survey

The field survey was conducted in the Medes Islands, NW Mediterranean (40°02′55″N, 3°13′30″E). Sampling and observations were carried out at 18–20 m depth within a coralligenous community located in a channel. The channel was alternately influenced by northerly and southerly currents, which may reach high speeds (from 2 up to 30 cm s−1, Rossi & Gili, 2007).

C. rubrum polyp expansion was monitored at a high frequency (i.e., once every 6 h) by a single SCUBA diver from June 24 to 29, 1997. This period was chosen because pelagic primary production and the frequency of seston pulses is high (Rossi & Gili, 2007). Expansion is defined as the maximum aperture of polyps (Sebens, 1987). Polyp activity was observed in ten groups of ten colonies each time by scuba divers.

The following parameters were concomitantly monitored: (1) hydrodynamics (using an Aandera SDP® Doppler current meter, moored in the same place as the observed passive suspension feeders, recording currents 0.5 m above the benthic surface), (2) seston concentration and quality determined by assessment of chlorophyll a and protein concentrations (see Rossi & Gili, 2007), and (3) zooplankton concentration determined by analysing two samples collected by a scuba diver towing two small plankton nets (22 cm in diameter with a mesh size of 100 µm) a distance of 40 m (Coma et al., 1994; Rossi et al., 2004). Wind, wave height and tidal oscillation were recorded every day by the Estartit Meteorological Station in accordance with the protocols of Cebrián, Duarte & Pascual (1996).

Experimental observations on polyp activity

On March 15, 2007, several small colonies of C. rubrum were collected at depths of 28–30 m (water temperature 14 °C) using a crowbar, and were immediately transferred to the Observatoire Océanologique de Banyuls (France) (see a description of the area in Rossi et al. (2003); this area is 40 km from the Medes Islands, where we made the field study). Colonies were placed in small cylinders, 4 cm in diameter (two colonies per cylinder), attaching them with non-toxic rubber, and kept at 15 °C in running seawater for two weeks prior to the start of the experiments. Corals were fed three times a week with copepods, ground mussels and Artemia nauplii. Running seawater supplied to the tanks was filtered through a 4 µm filter, allowing only pico- and nano-plankton to be present in the aquaria, which is a negligible part of the diet of these species (Tsounis et al., 2006; Picciano & Ferrier-Pagès, 2007).

The flume used (Fig. 1) was 10 cm wide × 20 cm high with a total length of 450 cm, resulting in an overall maximum water capacity of 85 litres. For each experiment, filtered (2 µm) seawater was used. Four colonies (two cylinders) were tested in each experiment. The pump used could generate speeds ranging from 1 to 6 cm s−1. A Minilab SD12 ultrasonic current meter (Sensordata, Bergen, Norway) (resolution 1 mm s−1, bandwidth 35 Hz and effective acoustic path length 29 mm) was used to measure water flow during each experiment. Water temperature was recorded to the nearest 0.5 °C.

Figure 1 The flume used for laboratory experiments.

It consisted of a closed transparent plastic ellipsoid channel placed in a temperature-controlled chamber.

The polyps of all colonies were closed at the beginning of each experiment. Colonies were placed in the flume and acclimated for 30 min, with experiments conducted from 24 to 72 h.

For assessment of the effects of zooplankton and nutritional stimuli, natural Mediterranean zooplankton was used.

Zooplankton was collected from epibenthic waters using 200 µm mesh nets near the coast on the 6th and 14th April, 2007. The samples were transported in a cooler to the main lab. The zooplankton was centrifuged (3,000 rpm) and stored at −20 °C until its use in the experimental set-up. An aliquot of 1 ml in 5 replicates from each sampling was fixed with 6% formalin in order to count the number of items added to the flume water in each experiment (Coma et al., 1994).

For the chemical signal experiments, a selected volume of zooplankton (40 and 120 ml in 65 litres, corresponding to nutritional stimuli N1 and N2) was gently ground with a glass homogeniser, in order to make a uniform mass of the zooplankton. The homogenised zooplankton was filtered through a 10 µm mesh to remove the solid part and the liquid part was stored 60–90 min before the experiment in a cooler. The liquid part was the nutritional stimulus added to the flume. A selected volume (120 ml in 65 litres, final concentration 1,500 ± 252 ind. m−3) was used directly in the experiment to understand the influence of zooplankton (not ground, the dead particles directly added to the flume) on polyp expansion (N3). In both cases, the zooplankton (filtrate or particles) was placed directly in the flume once the water was running, not before. All the nutritional stimuli experiments were conducted at a temperature of 18 °C with a constant flow speed of 3 cm s−1.

The flume was illuminated using LEDs (Lunartec 48 LED white 40W bulbs). A mirror oriented 45° with respect to the flume’s main axis was placed downstream from the monitored colonies so that they could be observed through cameras without major disturbance to the water current (Fig. 1). A video system with a colour camera (JAI S3200® fitted with a 50 mm objective) was placed next to the flume and used to monitor the colonies. The signal was received using a Falcon Plus® video grabber and transferred to a PC, where the images were recorded in JPEG format (80% compression). Real image size was 104 × 78 mm (Duchêne et al., 2000; Maire et al., 2007) corresponding to 736 × 568 pixels and thus to a resolution of 140 µm pixel−1. The frame capture rate was set at 3 images/min.

Maximum polyp expansion was recorded and calculated (taking into account the contracted and fully expanded polyp). The coral’s white polyps and tentacle crowns contrast with the black background and red coenenchima, allowing for good image analysis. The JPEG images were assembled into AVI films (SEM VIDEO 1). Image processing was then performed on the films using CVAB software (©J.C. Duchêne). Image analysis allowed calculation of the surface area of open polyps on the coral branches. The surface area of open polyps on each branch was separated and accounted for in every image of each film. Segmentation of the images allowed the open polyps to be separated from the coenenchyme in distinct pixel patches corresponding to existing regions of the coral. The labelled regions were tracked across images in the films, providing information on the activity of the polyps forming the colonies. Two types of information were derived from the observations: (1) the total surface area of the open polyps, and (2) the polyp activity index (taking into account the minimum and maximum expansion of the polyp, as total pixel counting), obtained from subsequent image differentiation. The polyp expansion showed the changes occurring in each region at any given time, including opening and closing polyps and moving tentacles. If a polyp moved its tentacles, the recorded surface area is expected to remain the same, with a null difference between the two successive apparent surface areas. While measuring colony size, the software also allowed to measure the events characterised by low dynamism, such as slow feeding movements, and examination of mesoglea inflation before polyp opening.

Statistical analysis

The variability in (i) C. rubrum polyp expansion, (ii) seston composition and (iii) zooplankton composition was assessed at various temporal scales by multivariate analyses and correlated with the environmental variables recorded during the sampling. By means of two laboratory experiments, we assessed the variability in C. rubrum polyp expansion in response to a range of temperatures, current speeds and nutritional stimuli.

In situ, to assess differences in C. rubrum polyp expansion, the design incorporated two factors: Cycle (Cy, as a fixed factor with 5 levels, each 24 h) and Time (Ti, as a random factor with 4 levels, nested in Cycle, each 6 h), with n = 3. Multivariate analyses of variance (PERMANOVA, Anderson, 2001) considered Euclidean distances based on untransformed polyp expansion data and previously normalised seston composition, using 9,999 random permutations of the appropriate units (Anderson & Braak, 2003).

To assess differences in zooplankton composition we performed permutational analyses of variance (PERMANOVA, Anderson, 2001) considering Bray Curtis dissimilarities based on transformed data (fourth root), using 9,999 random permutations of the appropriate units (Anderson & Braak, 2003), adopting a design with one factor, i.e., Cycle (Cy, as a fixed factor with 5 levels, n = 4). In order to detect which taxa contributed most to dissimilarity among the cycles, a similarity percentage (SIMPER) analysis was performed (Clarke, 1993). To examine the generality of patterns in polyp expansion, seston composition and zooplankton composition, we generated MDS plots.

In addition, we performed another two laboratory experiments in order to evaluate C. rubrum polyp expansion under a range of physical conditions and nutritional stimuli. To assess the effect of temperature and current speed on polyp activity, we performed permutational analyses of variance (PERMANOVA, Anderson, 2001) considering Euclidean distances based on untransformed data, using 9,999 random permutations of the appropriate units (Anderson & Braak, 2003), adopting a design with two factors: Temperature (Te, as a fixed factor with 3 levels) and Current (Cu, as a fixed factor with 3 levels) with n = 12.

Moreover we performed a further experiment to evaluate the effects of nutrient levels, following a design with one factor, i.e., Nutritional Stimuli (Nu, as a fixed factor with 4 levels) with n = 8.

When significant differences were encountered (p < 0.05), post-hoc pairwise tests were carried out to ascertain the consistency of the differences across the conditions tested. Because of the restricted number of unique permutations in the pairwise tests, p values were obtained from Monte Carlo samplings. The analyses were performed using PRIMER v. 6 (Clarke & Gorley, 2006).

Results

Polyp activity in high time-resolution field monitoring

The activity rhythms (polyp expansion) of C. rubrum between the 24th of June (15:00) and the 29th of June (9:00) 1997 in relation to the environmental variables tested are shown in (Figs. 2A to 2D). The activity of the 100 colonies varied between 0% (polyps fully closed) and 100% (polyps fully open), the change occurring in only 6 h in some cases (see for example the transition between 26 of June at 9:00 and 26 of June at 15:00). Current speed ranged from 0.2 cm s−1 to 30 cm s−1. The mean current speed during the activity rhythm observations was 9.3 ± 9.4 SD cm s−1, with the highest speeds recorded towards the middle and end of the experimental period (Fig. 2A). Zooplankton concentration (mainly copepods and nauplii) varied from 298 individuals m−3 to 8437 individuals m−3. The mean concentration was 2,122 ± 2,412 SD individuals m−3. Zooplankton had higher concentrations in the later cycles (Fig. 2B). Chlorophyll a concentration varied from 0.28 µg L−1 to 0.70 µg L−1, with a mean of 0.4 ± 0.1 SD µg L−1, with the highest values recorded at the beginning of the experimental period (Fig. 2C). Protein concentration followed a different tendency (Fig. 2D), ranging from 135 µg L−1 to 243 µg L−1. The mean concentration was 176 ± 32 SD µg L−1, with the highest concentrations found towards the middle and end of the experimental period. No significant relationships were directly observed between the tested environmental variables and red coral polyp expansion.

Figure 2 The activity rhythms (polyp expansion) of C. rubrum in relation to the environmental variables.

(A) Current speed, (B) zooplankton concentration, (C) chlorophyll a concentration, (D) protein concentration.

The results of the PERMANOVA reveal that the percentage of expanded C. rubrum polyps varied significantly among sampling times of cycles (Table S1), while pairwise analyses underline the differences within each cycle (Table S2).

Following the same experimental design, PERMANOVA analyses on seston composition exhibited significant differences at the investigated temporal scales (Tables S1 and S2). In particular, pairwise analyses of both polyp expansion and seston composition showed significant differences among all sampling times of the first cycle.

MSD plots of seston composition in relation to each sampling time show separation of cycles and sampling times (Fig. 3A), indicating the high number of open polyps during the cycles characterized by higher sea water movement. Significant differences among cycles were confirmed by Permanova for zooplankton composition (Table S3).

Figure 3 MDS plots for seston composition with Euclidean distances based on normalised data; vectors with Pearson’s correlation coefficients show (A) environmental variables and (B) polyp activity.

The SIMPER analysis revealed the highest dissimilarity in the zooplankton assemblages, reaching 30.74% for C1 vs. C5, followed by C2 vs. C5 (30.12%), C1 vs. C4 (27.02%), C1 vs. C3 (25.09%) and C2 vs. C4 (25.06%) (Table S4), highlighting variation between the first and last cycles. The MSD plots confirm the separation of cycles (Fig. 3B).

Laboratory experiments

In the experiments, a clear rhythm appears in the opening events of the coral branches (Fig. S1 SEM). This rhythm was most dominant when the colony’s polyps are fully extended as could be observed in the records of individual polyp activity (Fig. S2 SEM).

The results of the PERMANOVA showed a significant Te × Cu interaction (Tables S5 and S6, Fig. 4), demonstrating that C. rubrum polyp expansion varied significantly among the tested temperatures and currents. Increased temperature led to a rapid expansion of C. rubrum expansion in still-water, however the simultaneous interactions between current and temperature resulted in fast polyp expansion when the current speed was maximum and the temperature was low (Fig. 4).

Figure 4 The influence of temperature and current on C. rubrum polyp expansion under a range of experimental conditions.

C0, still water; C1, 3 cm s−1; C2, 6 cm s−1. T1, 13 °C; T2, 18 °C; T3, 25 °C. Data are reported as mean values ± S.E.

Nutritional stimuli

The results of the PERMANOVA (Table S7) and pairwise analyses (Table S8) revealed that C. rubrum polyp expansion varied significantly among the different nutritional stimuli tested, except for N1 vs. N2 (Fig. 5). In particular, the analyses showed a rapid expansion of C. rubrum polyps when nutrition stimuli increased, reaching a maximal reaction when nutritional stimulus consisted of zooplankton (Fig. 5).

Figure 5 The influence of nutritional stimuli on C. rubrum polyp expansion under a range of experimental conditions.

N0, no stimulus; N1, 40 ml; N2, 120 ml; N3, zooplankton (aprox 1,500 ind m−3). Data are reported as mean values ± S.E.

Discussion

The present study represents a first step to provide new insights into the relationship between environmental-biological conditions and the capacity of passive suspension feeders to intercept pulse-like energy inputs. High-resolution observations of polyp activity in the field highlighted the complex combination of environmental variables linked to seawater movement even on the small scale.

Ex-situ high-resolution flume experiments showed that polyp expansion accelerates with current speed. In addition, the presence of nutritional stimuli, especially zooplankton, induces a clear response in C. rubrum polyp activity, confirming they are sensitive in detecting food availability.

The outcomes also contribute to our understanding of the biology and ecology of red coral. Corallium rubrum polyp expansion seems to be most affected by water temperature, as observed by Picciano & Ferrier-Pagès (2007), and by sea water movement as revealed here. Passive suspension feeding depends on current flow. Nevertheless, given constant seston concentrations, increasing current speed enhance filtration up to a maximum beyond which filtration no longer increased (Wildish & Kristmanson, 2005). Our in situ results suggest complex hydrodynamic conditions act in a complementary way to shape polyp activity. It is clear that different conditions were present in the first and second slot of the week. During the last days, a combination of water movement and seston concentration could be the key to understanding increased activity of C. rubrum. Water movement effects are essential to understanding plankton activity and concentration (Sebens & De Reimer, 1977; Palardy, Grottoli & Matthews, 2006). Increasing water movement will increase this plankton activity and concentrations which, in turn, will increase the number of red coral expanded colonies.

While the assessment of the effect of drag forces on polyp retention ability is beyond the scope of the current study, we found a clear relationship between polyp activity and current speed, similar to what has been reported in studies of other gorgonian species (Dai & Lin, 1993). The rigid structure of the CaCO3 skeleton of C. rubrum creates a highly inflexible structure that has a limited range of movement, unlike other highly flexible gorgonian passive suspension feeders used in previous studies (Dai & Lin, 1993). Highly flexible organisms may be able to minimise drag forces by altering colony shape and reducing projected colony area when exposed to increased flow (Vogel, 1996). Previous studies of scleractinian corals suggest that current velocity within colonies has an upper limit, or saturation velocity, which is dependent on colony morphology (Chamberlain & Graus, 1975). The current speeds used in our study have no dramatic effect on polyp shape and are considered optimal for polyp particle capture (Leversee, 1976; Sponaugle, 1991). An increase in current speed (from 0 to 6 cm s−1) therefore increases particle delivery to the polyps. Dai & Lin (1993) showed that Acanthogorgia vega had a broader spectrum of polyp efficiency (as capture rates) at flow speeds ranging from 0 to 15 cm s−1 than the other two species tested, probably due to its bushy shape. In fact, the effect of flow on particle capture by polyps is probably a general phenomenon among octocorals (Robbins & Shick, 1980; Patterson, 1991; Dai & Lin, 1993), with polyp capture efficiency falling as the Reynolds number increases. In the present study, C. rubrum polyp activity tends to increase with current speed, even in the absence of increased abundance of food.

Similar to previous studies of C. rubrum polyp activity, the current study found a significant relationship between temperature and polyp activity, with less polyp activity at high temperatures. Previati et al. (2010) showed a significant relationship between oxygen consumption and activity (open polyps) at 18–20 °C and a current of 1 cm s−1 (approx.): oxygen consumption and activity increased, but above this temperature oxygen consumption decreased. Studies of other octocorals follow the same trend of closing their polyps at higher temperatures and decreasing oxygen consumption to maintain a decreased metabolic rate in a quasi-dormant stage (Coma et al., 2002; Previati et al., 2010). Although the relationship between temperature and polyp activity might be the result of endogenous rhythms related to an internal clock (Previati et al., 2010), it seems that lack of water flow (decreased current speed) is a key factor in the spontaneous opening of polyps in the absence of external stimuli. In the present study, colonies also decreased their maximum opening frequency as temperature increased in still-water conditions. One hypothesis is that polyp expansion is needed for gas exchange and excretion, however the majority of colonies tend to remain closed as much as possible as temperature increases. When food availability in the water column is low, the increase in temperature and flow seems to increase the response of C. rubrum (maximum opening frequency), indicating a balance between the need for opening and the current stimulus. Differences between the still-water and current-speed experiments in terms of maximum opening frequency suggest current stimulus has a greater influence than temperature constraints. Anthozoans need to expand their polyps to favour breathing (Previati et al., 2010). Our results suggests that food acquisition (related with water movement) in these passive suspension feeders appears to have priority over gas exchange.

Of all the variables used to test for a response in polyp activity in the present study, the addition of zooplankton elicited the fastest response in C. rubrum colonies. C. rubrum is considered a passive suspension feeder capturing particulate organic matter (POM) from the surrounding environment (Tsounis et al., 2006). Other Mediterranean and tropical asymbiotic octocorals (gorgonians, soft corals) are able to capture POM (Ribes et al., 2003), small zooplankton (Coma et al., 1994; Rossi et al., 2004) and phytoplankton (Widdig & Schlichter, 2001). C. rubrum is also able to feed on bacterioplankton (pico- and nanoplankton) (Picciano & Ferrier-Pagès, 2007). A chemical or chemical/physical (zooplankton) stimulus caused a rapid response in terms of polyp activity, in some cases within a few seconds. The rapid response of polyp activity increases with temperature, but at higher food concentrations the response becomes even more rapid (Grémare et al., 2004). Relying completely on heterotrophic inputs from seston, the detection of chemical signals and/or food particles may be more important than other variables (i.e., temperature or current). Although there may be inter-individual variability in the response (Duchêne et al., 2000; Duchêne, 2017), there is clearly less variability in polyp activity when zooplankton stimuli are combined with flow speed than with flow speed alone. This result is not surprising as it has been demonstrated in previous studies that the addition of food to the water column can elicit a response in other taxa (Duchêne & Rosenberg, 2001; Maire et al., 2007; Duchêne, 2017). The response to a chemical signal indicating increased zooplankton concentrations has not previously been experimentally tested in octocorals, but in scleractinian corals, particle concentration also elicited a response (Anthony, 1999). It is clear that the synergistic effects of higher current speed and the presence of zooplankton (or its chemical signal) provide a stimulus for the expansion of the polyps of this gorgonian. Therefore, we hypothesise that if food and current stimulate polyp activity, a recurring hydrodynamic parameter (food pulses due to high seston concentration) may cause current speed and particle concentration (dead or alive) to act synergistically.

The high-frequency in situ monitoring used in this study is currently the only known method for detecting the response in terms of polyp activity to changes in zooplankton and particulate organic matter availability and current flow speeds. In the space of just a few hours, epibenthic seston concentrations may fluctuate dramatically, with large increases and decreases in the concentrations of available zooplankton or seston (Rossi & Gili, 2007; Rossi et al., 2013). A greater frequency of high-speed current episodes may have a synergistic effect on the entire coralligenous community, by both increasing currents and resuspending particulate organic matter. This environment creates optimal conditions for nutrient cycling and capture of crustacean zooplankton and increasing prey capture rates among benthic suspension feeders. A relationship between food pulses and feeding activity was also found in other tide-dominated environments (Naylor, 1976; Naylor, 2005).

We consider that in the Mediterranean Sea (and in other benthic systems), food availability is non-continuous for benthic suspension feeders. Increasing the frequency of high current-speed events and hence the quantity of available epibenthic seston may be a driver of pulse-like temporal changes in the particulate organic matter available in the water column for the energy budgets of coralligenous (and other) benthic communities. Many authors have shown the positive relationship between prey capture rates and concentrations of plankton (Sebens & De Reimer, 1977; Coma et al., 1994; Palardy, Grottoli & Matthews, 2006). In intertidal systems, there is a clear relationship between benthic suspension feeding activity and tidal fluxes (Sebens, 1987). In C. rubrum, short periods of high seston and zooplankton abundance could be the key to understanding energy input, high-current episodes creating high prey concentrations, leading to maximum particle capture rates.

Optimal foraging theory (Hughes, 1980) posits the need to take advantage of favourable feeding pulses as an individual colony but also as a population within a community. Palardy, Grottoli & Matthews (2006) suggested that the energy budget of passive suspension feeders may be dependent on non-continuous zooplankton availability, and Robbins & Shick (1980) related the activity of Metridium senile to tidal flux. It is clear that even if polyp seston capture is an important source of nutrition (being a more constant food source, Ribes, Coma & Gili, 1999) the detected seasonal concentrations may not be sufficient, given the energy constraints of most passive suspension feeders. In the complex coralligenous community, a broad spectrum of energy constraints is shown by the diverse range of activities and behaviours observed during our study period. Many organisms take advantage of the food pulses related to tidal patterns of water movement, resuspension and nutrient recirculation (Robbins & Shick, 1980; Gibson, 2003). We hypothesise that the foraging strategy of C. rubrum (but also other benthic organisms and communities) is influenced by the frequency of high current-speed events, which are far from homogeneous in different seasons (Rossi & Gili, 2007).

Conclusions

In this paper we showed that temperature and current speed are essential cues to understand the polyp expansion of passive suspension feeders; however, chemical signals have a prevalence in the activity of these organisms. The synergy between currents and zooplankton is thus the key to understand prey capture of benthic suspension feeders in natural environments. Such a combination is not homogeneous through the time, so when we try to understand and quantify the energy balance of these metazoans, we have to consider the available food pulses that may be in phase with tidal rhythm and also affected by strong winds or temporal upwellings. The presence of food pulses is key to understanding global energy inputs and the energy budgets of these organisms, and how these synergistic effects (current speed with particle concentration) bring energy pulses to the benthic communities.

Supplemental Information

Figure S1 SEM: Periodograms from three different colonies

Example of three periodograms from three different colonies (peaks represent polyp expansion), showing endogenous rhythms at 18 °C and still-water conditions. On the left the recorded normalised activities (i.e., the number of pixels divided by the maximum polyp expansion for that experiment); on the right the Lomb periodogram with frequencies on the X axis and number of occurrences on the Y axis. Figures close to the peaks indicate the periods. The 3 dashed lines represent the significance of the peaks, 0.1, 0.01 and 0.001, the smallest value corresponding to the highest significance.

Click here for additional data file.

Figure S2 SEM: Records of individual polyp activity

(A) The area below the peaks for a given experiment. (B) The derivative of this curve with absolute values (increase or decrease in polyp expansion). These records usually show a steeper descent after opening.

Click here for additional data file.

Video S1 Video: Corallium rubrum polyp activity at 18 °C and 3 cm s_-1 current speed

Click here for additional data file.

Supplemental Information 4 Polyp expansion of passive suspension feeders: a red coral case study, supplementary tables

Tables S1–S8: tables with the different statistical analysis.

Click here for additional data file.

Supplemental Information 5 Raw Data Corallium rubrum: senton characteristics, polyp expansion, zooplankton concentration, and results in different experiments as explained in the main text

Data of the (1) field observations (seston and zooplankton) and (2) experiment variables (temperature, currents & chemical signals)

Click here for additional data file.

The authors are especially grateful to Josep Pascual for environmental data collection and preliminary data processing, and to Josep-María Gili, Rafael Coma, Bernat Hereu, Marta Ribes, David Diaz, Marc Marí and Mikel Zabala for field assistance. Imma Llobet and Elisabetta Broglio made an important contribution to the zooplankton counting. We also thank Prof. Antoine Grémare for laboratory facilities. We also thank Andrea Gori and Darren Brown for their useful comments on the manuscript.

Additional Information and Declarations

Competing Interests

Author Contributions

Data Availability

The authors declare there are no competing interests.

Sergio Rossi conceived and designed the experiments, performed the experiments, analyzed the data, contributed reagents/materials/analysis tools, prepared figures and/or tables, authored or reviewed drafts of the paper, approved the final draft.

Lucia Rizzo analyzed the data, prepared figures and/or tables, authored or reviewed drafts of the paper, approved the final draft.

Jean-Claude Duchêne conceived and designed the experiments, analyzed the data, contributed reagents/materials/analysis tools, prepared figures and/or tables, authored or reviewed drafts of the paper, approved the final draft.

The following information was supplied regarding data availability:

Raw data is available in the Supplemental Materials.

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
