# Peer review of "Polyp expansion of passive suspension feeders: a red coral case study"

_PeerJ, doi:10.7717/peerj.7076_

## Round 0.1 · original submission · Major Revisions

Two expert reviewers have evaluated your manuscript and thir comments can be seen below. Both reviewers agree that this is an interesting contribution backed by a powerful data set and will be an important addition to the literature on the topic of suspension feeders.
However, both reviewers remarked that in general the manuscript was confusing. More details were needed in the description of the experimental design and this section needs to be explained more clearly, the results need to be more clearly reported and the number of tables reduced and synthesized. My suggestion would be to do this and to move the tables to the supplements section. The Discussion also needs work to be restructured, reorganized and reduced in length. You may also want to consider reworking the figures.

Reviewer 1 ·

Basic reporting

The topic of the paper is highly interesting for biologists and ecologist studying suspension feeders. Several different approaches are applied to highlight the effect of flow speed and seston concentration on the espantion cycle of Corallium rubrum (an high valuable octocoral) polyps. Also if authos collected a wide bulk of data, however the manuscript is confused in some parts and Results must be reconsidered and more clearly explained and Discussion must be reorganised and condensed.

Experimental design

Also if a lage amount of data have been collected the general experimental design should be more clearly explained and synthesized.

Validity of the findings

Some main findings are valid and important, however they must be more clearly reported, explained and discussed. Tables should be reduced and must be more synthetically sketched.

Some of the points of the manuscript that must be readdressed are reported in the following Comments for the author.

Additional comments

I suggest to the authors to expand the Results section and to describe in a more detailed way the main results that must be more clearly highlighted.
The number of tables should be reduced and they must be condensed.

Some main points that should be readdressed:

Abstract

line 26 "polyp activity correlated positively with particle concentration, which was related to current speed.." - polyp activity which was significantly (?), positivelty correlated to particle concentration, which, in turn, was singnficanly correlated to current speed ? Is this the meaning of this sentence?

lines 28-31 "....which showed differences in polyp activity.......but especially in response of nutritional stimuli and the presence of zooplankton. Zooplankton and water moviment..." - Differences means oscillations? presence means density? Zooplankton density? Whater speed? this sentence must be write again.

Material and methods

lines 123- 175 - This part may be condensed.

line 176-177 "..2) the polyp activity index, obtained from subsequent image differentiation. - This sentence is not clear, please describe this index in a more detailed way (formula?).


Introduction

line 54 "..is to change their feeding activity.." - to tune their feedng activity?

Results.

Results must be more exhaustively and clearly written and the number of subdivisions must be reduced (Laboratory experiments is made of 3 lines only..)

242-244 " No significant relationships were directly observed between the tested environmental variables and red coral polyp expansion" - This is main point of the Results that must be explained and deeply discussed in the Discussion. "Tested variables" = recorded variables.

Line 251 MDS plots of seston cmposition........show separation of sycles and sampling times" - Indicating or suggesting what?


Line 264 "..moviments increase until the colony suddently closes” – This sentence must be written in a more clear way.

Line 269 “Pairwise analyses clarified the significance of individual comparisons “ – As the Interactions in this ANOVA are significant, no further analysis can be done.

Discussion

278 “The present study provides new insights into…” - This study try to provied some new insights into…

292 “ Our in situ results suggest complex hydrodynamics conditions act in a complementary way….”
- Go deeper, discuss in a more eshaustive way.

295 “polyp efficiency” – polyp activity?

323-326 –This sentence is not clear and must be written in a more plan way.

360 “…presence of naural prey..” – Density?

375 “The relationship between food pulses and feeding activity has also been studied in other ….”
– A relationship…….was also found in…

382 “ Many authors have shown the relationship between..” – Positive relationship between…

405 “ ....bring energy pulses to the bentic community which may change our view of the energy budgets of benthic communities”. – bring energy pulses to the benthic communities. (end here this sentence).

Some minor points

Abstract
line 17 "has been linked" - Has been cosidered to be affected by....
line 22 " testing various ecological varables" - please change in: examining...

Material and method
line 96 C.rubrum polyp expansion was monitored at high frequency.." - by a video camera....
line 101 "Several environmantal and biological parameters..." - The following parameters..

189 and 192 - "The differences in.. " - Variability in...
198 "seston data" - seston density? composition?
377 “ We hypothesize…” – We consider that…

Results
246 "Polyp expansion.." - Polyp expanson rate? Percentage of expanded polyps?

Discussion
282 “even on the small scale. “ – End the line here and start the new sentence on a new line.

Reviewer 2 ·

Basic reporting

The manuscript prepared by Rossi and colleagues is written in professional English and easy to understand.

General:

1. What were the temperature and currents speed used in the experiment? And what are the different nutritional stimuli? I believe this is very important information, so I do not understand why this is not mentioned in the text.

2. The wording is precise though some times very specific so readers from a slightly different field might be confused. To make it easier, the terms 'seston' and/or 'flume' could be explained in the introduction.

3. I appreciate the amount of statistical detail given in the results section, however, the numbers break up the flow and make it rather difficult to follow the chain of thoughts. May be the authors would like to only highlight the key numbers of the statistics.

4. The manuscript contains a lot of figures. Figure 2 is very difficult to read in its current preparation, may be a landscape type of outline would be easier or only focus on the graphs that are really important for the understanding of the manuscript. Figures 3 and 4 are very similar though focusing on different aspects of the experiment (seston and zooplankton, respectively). Can they however be combined? For figures 5 and 6 I am missing information in the figure caption on the temperature and current conditions as well as the nutritional stimuli.

5. The tables are very repetitive and should contain more detail as mentioned for the figures. In table 4 TexCu reads as one word, Te x Cu would increase the readibility.
I believe that the caption for figure 7 is the one for figure 5.

6. The discussion is very long and some of the information is a bit confusing. I recommend to condense this section and focus on the main aspects. Further I would appreciate more clearly separated paragraphs this would increase the readibility.
A good idea might be to first discuss the two experiments separately and then have a concluding section.

Minor:
Line 166: I believe there is a typo 'coenenchyme' instead of 'coenenchima'.

Line 288: Does 'and by seawater movement' refer to the current study/manuscript?

Line 295: 'polyp efficiency' is explained later in the text as particle caputre but may be already explain it here.

Line 358: Is Anthony (1999) the only paper? Otherwise the sentence could be rephrased as followed 'The response to a chemical signal indicating increased zooplankton concentrations has not previously been experimentally tested in octocorals, but in scleractinian corals particle concentration also elicited a response (Anthony, 1999).

Line 577 and 578: May be 'significance' is a better term instead of 'significativity'

Experimental design

The experimental design as comparison of field survey and experimental observations is a great approach to validate field observations under laboratory controlled conditions.
However the description appears to be somewhat confusing to me. Therefore, I recommend to revise this section and may be add some more details.
Further, I do miss some more detailled informations about the design:

1. Can you give more information about the sampling locations, e.g. were the colonies for the experimental observation collected at the same location as the field survey was conducted? Further was the temperature also recorded as part of the environmental parameters?

2. How far apart were the groups of colonies? What type of light was used for the night observations? Could that have influenced the opening/closing of the polyps?

4. What type of axe (line 114) was used for the colony collection? This is more a 'out of curiosity question' as I haven't used an axe for coral sampling before.

5. What light conditions were used in the culture tanks and the flume?

6. Would it be possible to describe the plankton addition in more detail? I do not quite understand the different volumes used for homogenisation (line 146) and what was added.

7. The statistical analysis section is very well described!

Line 106: '..., determined by analysing two samples...' Does this mean two samples at each group of colonies of two samples during each dive?

Lines 134-136: What does it mean 'acclimated for 30min' and 'experiments conducted from 24 to 72 hours'?

Validity of the findings

As mentioned above, the results and discussion should be revised. The authors present interesting results which could be highlighted more. The conclusions are valid with the findings however should be more precise and condense.
Some of the information given in the discussion could already be highlighted in the introduction which would specialise the paper more.

Additional comments

The manuscript 'Poylp expansion of passive suspension feeders: a red coral case study' is well written and will contribute to its respective area of research. Prior to publication, I recommend careful revision of the manuscript with some improvments to increase the readibility and to make it more attractive for researchs with other backgrounds.

---

## Round 0.2 · Minor Revisions

Two expert reviewers have evaluated the revised version of your manuscript. Both agree that the manuscript has improved considerably and that it is almost ready to be accepted. However, both have found a number of typographical errors and details that need to be corrected.

My recommendation is that these errors and details be attended to and the manuscript be resubmitted as soon as possible.

Reviewer 1 ·

Basic reporting

The manuscript has been improved and is basically ready for publication. I only suggest some minor change. Moreover I suggest to the authors that Discussion should be condensed.
Minor changes are reported in the following.

Experimental design

This was examined in my previous revision.

Validity of the findings

This was reported in my previous revision.

Additional comments

The manuscript has been improved and is basically ready for publication. I only suggest some minor changes. Moreover I suggest to the authors to condense a little the Discussion section.

Some minor changes:

Abstract
line 25 - please, delete comma.

Material and Metods
line 113 - his sentence is not clear, do you mean that a similar,previosly described area is 40 km apart from the sampling site? If so you can delete the sentence or explain that a similar area (40 km part) was described by.....

Still water - Do you mean marine natural water or marine filtered or stelirized water or what?

161 maximum poyp expansion - Is it the maximal percentage of polyps expanded in a colony or what? Please, expain in a detalied way.

172 Polyp activity index - Please, explain more clearly this index and add a formula. You have to explain widely these indexes so that they can be appied by other researchers in the future ofcomparison....

Results

236 PEMANOVA - So what is the meaning of this result? Can you assess that an higher variation occurs between cycles or between times? Should you go deeper in this result?

239 Significant differences among temporal scales? Is there some differences between scales?

243 - OK
244 on higher water moviment?
249 - OK
254-255 ...so what? this sentence must be completed...
256 significant Te x Cu term - significant Te x Cu interaction
258-258 increase of temperature support a rapid expansion of C. r expansion - to much expansions, the first one should be: increase..
254 simultaneous effects - interactions?

Discussion
line 278 confirming its capacity to detect....- to be sensitive to food availability
280 the obtained outcomes contribute....-try to contribute..
290 Water moviment is essential to understand..- the knowledge of water moviment effects is essential to understand...
296 has been seen in studies..- was reported in ....
404 but the presence of chemical signals have...- however, chemical signals have....
409 ta may be in consonance..- that may be in phase with tydal rhythm and be also affected by storms and upwellings

Reviewer 2 ·

Basic reporting

The submitted version by Rossi and coauthors has been extensively revised and the reviewers comments have been answered in great detail.

However, I found a few typos and have some suggestions for rewording to increase the clarity of the manuscript:

L30: '...stimulus consisting in zooplankton'. My suggestion would be '...of zooplankton'
L53: Please replace 'suspension-feeders' by 'suspension feeders'
L77: I would suggest '...using the red coral...'

L139: Replace pending with until '... and stored at -20°C until the experimental setup'
L145: '... uniform mass of the zooplankton'
L145-146: 'The homogenate was filtered through a 10µm mesh to remove the solid part and ...' might be eaiser to undestand
L150: Please use ground instead of grinded
L169: Do you mean calculation when saying computation?
L181: '... the software also allowed to measure...'
L187: '... was assessed...'
L212: '.... we performed another experiment...'
L242: '... pairwise analyses revealed...'

L260: May be rephrase to if it's correct 'This rhythm was most dominant when the colony's polyps are fully extended as could be observed in the records of individual polyp activity'
L272: '...the analyses showed...'

L322: '...with the polyp activity being lower...'
L324: '...current speed...'
L325: Above which temperature did the oxygen consumption decrease, 20°C?
L378-382: Can you please break this phrase into two. This would greatly increase the understanding of this sentence.

Experimental design

The experimental design and descripton of methods was revised according to reviewers comments.

Validity of the findings

I appreciate the revisions on the discussion part of this manuscript. The new version is more readable and easier to follow.

I do have one more general question: How can you differentiate between opening for food and opening for gas exchange? (L342-344)

Additional comments

Rossi and coauthors made great improvements on the manuscript. I appreciate the great detail with which reviwers' suggestions and comments were answered.

---

## Round 0.3 · Minor Revisions

I am satisfied with the response to reviewers and the modifications made to the manuscript. I Have read through the manuscript and made some editorial suggestions to help with the readability of the finished manuscript. Please consider these suggestions and incorporate them if you agree with them.

---

## Round 0.4 · accepted · Accept

I am satisfied with the changes made to the manuscript.